# Plant DNA polymerases α and δ mediate replication of geminiviruses

Mengshi Wu [1,2,3], Hua Wei [1,3], Huang Tan [1,3], Shaojun Pan[1,3], Qi Liu [2], Eduardo R. Bejarano [4] & Rosa Lozano-Durán [1,5 ✉]

Geminiviruses are causal agents of devastating diseases in crops. Geminiviruses have circular single-stranded (ss) DNA genomes that are replicated in the nucleus of the infected plant cell through double-stranded (ds) DNA intermediates by the plant DNA replication machinery. Which host DNA polymerase mediates geminiviral multiplication, however, has so far remained elusive. Here, we show that subunits of the nuclear replicative DNA polymerases α and δ physically interact with the geminivirus-encoded replication enhancer protein, C3, and that these polymerases are required for viral replication. Our results suggest that, while DNA polymerase α is essential to generate the viral dsDNA intermediate, DNA polymerase δ mediates the synthesis of new copies of the geminiviral ssDNA genome, and that the virus-encoded C3 may act selectively, recruiting DNA polymerase δ over ε to favour productive replication.

[1] Shanghai Center for Plant Stress Biology, CAS Center for Excellence in Molecular Plant Sciences, Chinese Academy of Sciences, Shanghai, China.
[2] Bioinformatics Department, School of Life Sciences and Technology, Tongji University, Shanghai, China. [3] University of the Chinese Academy of Sciences, Beijing, China. [4] Instituto de Hortofruticultura Subtropical y Mediterránea "La Mayora" (IHSM-UMA-CSIC), Area de Genética, Facultad de Ciencias, Universidad de Málaga, Campus de Teatinos s/n, Málaga, Spain. [5] Department of Plant Biochemistry, Centre for Plant Molecular Biology (ZMBP), Eberhard Karls University, Tübingen, Germany. ✉email: rosa.lozano-duran@zmbp.uni-tuebingen.de

Being obligate intracellular parasites, viruses rely on the host molecular machinery to replicate and spread. Gemini-viruses are a family of plant viruses with circular single-stranded (ss) DNA genomes, causal agents of devastating diseases in crops worldwide (reviewed in refs. [1,2]). None of the geminivirus-encoded proteins is a DNA polymerase, and gemi-niviral replication, which occurs in the nuclei of infected cells, completely relies on the plant DNA replication machinery. In a first step, the viral ssDNA has to be converted into a double-stranded (ds) intermediate, which is then replicated by rolling-circle replication (RCR) and recombination-dependent replication (RDR), producing multiple copies of the original viral genome that are eventually encapsidated and can be transmitted by the insect vector (reviewed in refs. [3,4]). Only one viral protein, the replication-associated protein (Rep), is required for the replication of viral DNA: Rep reprograms the cell cycle, recruits the host DNA replication machinery to the viral genome, and mediates nicking and rejoining events required for the initiation of replication and release of newly synthesized molecules (reviewed in refs. [3,4]). Another viral protein, C3, plays an ancillary role in viral DNA replication, acting as an enhancer in this process through an as-yet-unknown mechanism, but for which homodimerization and interaction with Rep are required[5–13]. A few host factors interacting with Rep and/or C3 and potentially required for geminiviral DNA replication, including the sliding clamp proliferating cell nuclear antigen (PCNA), the sliding clamp loader replication factor C (RFC), and the ssDNA-binding protein replication protein A (RPA), have been described to date ([9,14–16]; reviewed in ref. [4]); a recent genetic screen has identified a number of factors required for geminivirus replication in yeast, which can act as a surrogate system[17]. However, so far, no DNA polymerase associated with these viral proteins has been identified, although their activity is *conditio sine qua non* for geminiviral multiplication. A contribution of translesion DNA polymerases to the replication of geminiviral DNA has been proposed, but they were nevertheless not found essential for this process[18]. To date, the identity of the plant DNA polymerase replicating the viral genome has remained elusive.

Here, we show that the regulatory subunits of the nuclear replicative DNA polymerases α and δ physically associate with the geminivirus-encoded replication enhancer protein, C3, and that the activity of these polymerases is essential for viral repli-cation. Our results indicate that DNA polymerase α is required for the generation of the viral dsDNA replication intermediate, while DNA polymerase δ is involved in the downstream accu-mulation of newly synthesized ssDNA. In stark contrast with the other two replicative DNA polymerases, DNA polymerase ε exerts a negative effect on viral DNA replication. Taken together, the results presented here suggest a model according to which the viral C3 protein may act selectively mediating the productive recruitment of DNA polymerase δ over ε to enhance geminivirus replication.

## Results

**The geminivirus-encoded C3 protein interacts with POLA2, a subunit of DNA polymerase α, which is required for gemini-virus replication.** In order to identify host factors involved in the replication of geminiviral DNA, we performed a yeast two-hybrid (Y2H) screen using C3 from *Tomato yellow leaf curl virus* (TYLCV, genus *Begomovirus*) as bait against a cDNA library from infected tomato plants[19]. Interestingly, we found that C3 interacts with the N-terminal part of DNA polymerase α subunit 2 (SlPOLA2), the regulatory subunit of this holoenzyme (Fig. 1a); this interaction was confirmed in yeast and in planta through Y2H, co-immunoprecipitation (co-IP), and bimolecular

fluorescence complementation (BiFC) assays, and could also be detected with POLA2 from the *Solanaceae* experimental host *Nicotiana benthamiana* (NbPOLA2) (Fig. 1b–d; see "Methods"; Supplementary Table 1). In addition, SlPOLA2 and NbPOLA2 interact with the C3 protein encoded by the geminiviruses *Beet curly top virus* (BCTV, genus *Curtovirus*) and *Tomato golden mosaic virus* (TGMV, genus *Begomovirus*) (Fig. 1c, d and Sup-plementary Fig. 1), suggesting conservation of this interaction at least within the genera *Begomovirus* and *Curtovirus*. Of note, TYLCV induced the formation of nuclear speckles by POLA2 (Fig. 1e and Supplementary Fig. 2), although the functional relevance of these nuclear bodies remains to be determined. Chromatin immunoprecipitation (ChIP) assays indicated that NbPOLA2 can bind the viral genome; since no significant dif-ferences are detected between the wild-type (WT) and a C3 null mutant virus, we conclude that this binding occurs in a C3-independent manner (Fig. 1f and Supplementary Fig. 3). Knocking down *NbPOLA2* by *Tobacco rattle virus* (TRV)-medi-ated virus-induced gene silencing (VIGS) in *N. benthamiana* rendered plants with reduced height and thicker leaf blades (Supplementary Fig. 4). Strikingly, although *POLA2* silencing did not impair *Agrobacterium tumefaciens*-mediated transient trans-formation (Supplementary Fig. 5), it almost completely abolished local TYLCV replication and systemic infection (Fig. 1g, h), indicating an essential role of POLA2/DNA polymerase α in the replication of the viral genome. A similar effect of *NbPOLA2* silencing was observed on BCTV replication (Supplementary Fig. 6), suggesting that the role of this polymerase in viral repli-cation is likely conserved across different geminivirus species. Silencing of the gene encoding the catalytic subunit of DNA polymerase α, *POLA1/ICU2*, had a similar inhibitory effect on the accumulation of TYLCV (Fig. 1i), supporting the potential replicative function of this polymerase on the viral DNA, and ruling out a specific effect of the POLA2 subunit.

**DNA polymerase δ, but not DNA polymerases ε or ζ, is required for geminivirus replication.** In eukaryotes, DNA polymerase α primes DNA replication, while DNA polymerases δ and ε act as the main processive replicative polymerases. These three replicative polymerases are assembled into a large complex termed replisome, which contains all proteins required for DNA replication, including PCNA, RFC, and RPA (reviewed in ref. [20]). In yeast, DNA polymerase δ elongates the RNA/DNA primers produced by DNA polymerase α on both strands[21] and, according to the generally accepted model, then synthesizes the lagging strand, with DNA polymerase ε synthesizing the leading strand[22]; in addition, DNA polymerase δ is believed to perform initiation and termination of replication on both strands[23]. Recently, an alternative model has been proposed, according to which DNA polymerase δ would replicate both strands of the DNA, and the switch to DNA polymerase ε would only occur following replication errors[24]. The DNA polymerase δ regulatory subunit NbPOLD2 associates with TYLCV C3 in co-IP experi-ments, and interacts with TYLCV, BCTV, and TGMV C3 in BiFC assays (Fig. 2a–c and Supplementary Fig. 1; see "Methods"; Supplementary Table 1); no interaction could be detected between C3 and the DNA polymerase ε regulatory subunit NbPOLE2/DPB2. Intriguingly, co-expression of NbPOLD2 and geminiviral C3 led to a dramatic change in the morphology of the fibrillarin-positive nuclear bodies (Fig. 2a). Despite this apparent difference in C3 binding, both NbPOLD2 and NbPOLE2/DPB2 can bind the viral genome in ChIP assays, as previously shown for POLA2 (Fig. 2d, e and Supplementary Fig. 7). Of note, lack of C3 in a TYLCV C3 null mutant enhances binding of NbPOLE2 to the viral DNA, while increasing variation in the binding of

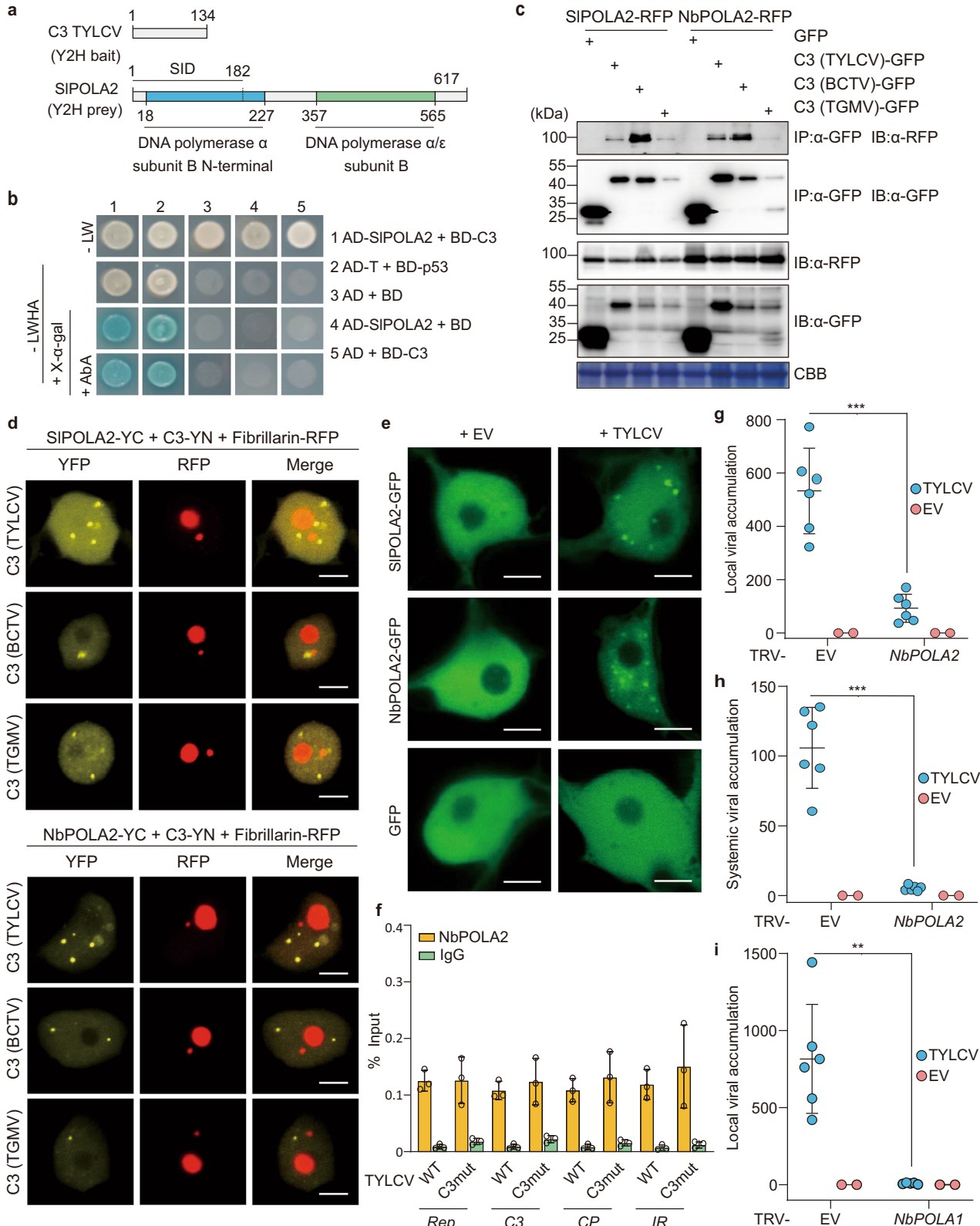

NbPOLD2 (Fig. 2d, e). With the aim to decipher whether DNA polymerase α acts in concert with DNA polymerases δ and/or ε in the replication of the viral DNA, we silenced the corresponding regulatory subunits *POLD2* and *POLE2/DPB2* by VIGS in *N. benthamiana*, and tested the capacity of TYLCV to replicate in local infection assays and to infect systemically. Silencing of either subunit results in a distinct developmental phenotype, with plants

of smaller size with leaves of abnormal shape (Supplementary Fig. 4), but the ability of *A. tumefaciens* to mediate transient transformation in these plants is not affected (Supplementary Fig. 5). Knocking down *POLD2* impaired TYLCV DNA replication in local infection assays (Fig. 2f), whereas knocking down *POLE2* enhanced viral accumulation both locally and systemically (Fig. 2g, h). The same effect could be observed for BCTV,

**Fig. 1 The geminivirus-encoded C3 protein interacts with POLA2, the regulatory subunit of DNA polymerase α, which is required for geminiviral replication. a** Schematic representation of bait (C3) and prey (SlPOLA2) isolated from the Y2H screen. Amino acid residues are indicated. SID: selected interaction domain. **b** C3 and POLA2 interact in yeast. AD: activation domain; BD: binding domain. The interaction between the SV40 large T antigen (T) and the tumor suppressor p53 is a positive control; empty AD- and BD-containing vectors (AD and BD, respectively) are used as a negative control. **c** C3-GFP (from TYLCV, BCTV, and TGMV) co-immunoprecipitates SlPOLA2-RFP (left) and NbPOLA2 (right) upon transient expression in *N. benthamiana*. IP: immunoprecipitate; IB: immunoblot; CBB: Coomassie brilliant blue. The predicted protein sizes are as follows: SlPOLA2-RFP, ~100 kDa; NbPOLA2-RFP, ~100 kDa; C3 (TYLCV)-GFP, ~42 kDa; C3 (BCTV)-GFP, ~42 kDa; C3 (TGMV)-GFP, ~42 kDa; GFP, ~26 kDa. Full blots and membranes can be found in the Source data file. **d** SlPOLA2 and NbPOLA2 interact with C3 from TYLCV, BCTV, and TGMV in BiFC assays upon transient expression in *N. benthamiana*. Fibrillarin-RFP marks the nucleolus and the Cajal body. Images were taken at 2 days post inoculation. Scale bar: 5 μm. Negative controls are shown in Supplementary Fig. 1. **e** Nuclear distribution of transiently expressed SlPOLA2-GFP, NbPOLA2-GFP, and free GFP in the absence (empty vector, EV) or presence of TYLCV in *N. benthamiana*. Scale bar: 5 μm. Additional images are shown in Supplementary Fig. 2. **f** NbPOLA2 binds the TYLCV genome in ChIP assays. The location of primers used for different genomic regions is shown in Supplementary Fig. 3a; the results for additional genomic regions are shown in Supplementary Fig. 3b. Data are the mean of three independent biological replicates; error bars indicate SD. *ITS* is used as the normalizer. **g–i** Viral accumulation in local (**g, i**; 3 days post inoculation) or systemic (**h**; 2 weeks post inoculation) TYLCV infections in *POLA2*-silenced (TRV-NbPOLA2) (**g, h**), *POLA1*-silenced (TRV-NbPOLA1) (**i**) or control (TRV) *N. benthamiana* plants measured by qPCR. Plants inoculated with the empty vector (EV) are used as a negative control. Data are the mean of six independent biological replicates; error bars represent SD. The 25S ribosomal DNA interspacer (*ITS*) was used as a reference gene; values are presented relative to *ITS*. The phenotype of silenced plants and silencing efficiency are presented in Supplementary Fig. 4. All experiments were repeated at least three times with similar results, with the exception of the ChIP assays, which were repeated twice. Asterisks indicate a statistically significant difference according to a two-sided Student's *t* test (***$P < 0.001$; **$P < 0.01$). The original data from all experiments and replicates can be found in the Source data file.

indicating that the role of DNA polymerases δ and ε in viral DNA replication is likely conserved in different geminivirus species (Supplementary Fig. 6b, c). Supporting the role of the DNA polymerase holoenzymes, and not specifically of POLD2/POLE2, silencing of the genes encoding their respective catalytic subunits, *POLD1* and *POLE1*, had similar effects on viral accumulation to those observed when silencing the regulatory subunits (Fig. 2i–k). While in *POLE2*-silenced plants the accumulation of *POLA2* and *POLD2* transcripts was decreased, no consistent reduction of transcripts encoding subunits of DNA polymerase α was found upon *POLD1/POLD2* silencing, hence ruling out indirect effects based on changes in the availability of this polymerase (Supplementary Fig. 8).

In yeast and mammalian cells, the translesion DNA polymerase ζ shares two regulatory subunits with DNA polymerase δ, namely POLD2 and POLD3 [25,26]. In order to test whether the detected effect of silencing *POLD2* on viral DNA replication may derive from an impact on the activity of DNA polymerase ζ, we silenced the gene encoding the catalytic subunit of this complex, REV3 (Supplementary Table 1). As shown in Supplementary Fig. 9, *REV3* silencing did not affect viral accumulation in local infection assays, in sharp contrast to *POLD1* or *POLD2* silencing (Fig. 2f, i), indicating that DNA polymerase ζ is not required for geminiviral replication. Our results, therefore, point to DNA pol δ, but not DNA pol ε or DNA polymerase ζ, as required, together with DNA pol α, for replication of the geminiviral genome.

**DNA polymerase α is required for the synthesis of the viral dsDNA replicative intermediate, while DNA polymerase δ is required for the downstream accumulation of ssDNA.** We next used two-step anchored qPCR[27] to quantify the relative accumulation of viral ssDNA (viral strand, VS) and the dsDNA intermediate (as complementary strand, CS) (Fig. 3a) in local infection assays with TYLCV-WT or a C3 null mutant in *N. benthamiana* plants in which *POLA2*, *POLD2*, or *POLE2* have been silenced by VIGS (Fig. 3b–e). In agreement with previous results, the lack of C3 resulted in a decrease in the accumulation of both strands of the viral DNA, consistent with the role of this viral protein as a replication enhancer (Fig. 3b–e;[9]). Strikingly, silencing of *POLA2* impaired the accumulation of the viral complementary strand (Fig. 3b, c), hence compromising the subsequent production of the viral strand, which requires dsDNA

as a template (Fig. 3d, e); silencing of *POLD2* did not affect the synthesis of the viral complementary strand, but interfered with the downstream accumulation of viral ssDNA (Fig. 3b–e), pointing at a function of this DNA polymerase in RCR. As previously observed (Fig. 2g), the lower levels of *POLE2* led to an increased accumulation of viral DNA, an effect that could be observed on both viral strands (Fig. 3b–e). Taken together, these results suggest that DNA polymerase α is essential for the initial synthesis of the viral complementary strand, and therefore for the generation of the dsDNA replicative intermediate, ultimately limiting the accumulation of both dsDNA and ssDNA, while DNA polymerase δ, alone or in combination with α, is required for the following RCR.

**The lack of C3 can be complemented by silencing of DNA polymerase ε subunits.** The finding that both DNA polymerases δ and ε can associate to the geminiviral genome, but δ is required for viral replication while ε seems to exert a negative effect that is released by its silencing, suggests that both DNA polymerases may compete for binding to the viral DNA with opposite outcomes, since only DNA polymerase δ leads to replication. The observations that (i) C3 can physically interact with POLD2, and (ii) the non-productive binding of POLE2 to the viral DNA is increased in the absence of C3, hint at the possibility that C3 might mediate the selective recruitment of DNA polymerase δ over ε. Supporting this hypothesis, silencing of *POLE2* increases DNA replication of a C3 null mutant TYLCV to wild-type-like levels in control plants (Fig. 3d, e). In order to further test this idea, we inoculated *N. benthamiana POLE2*-silenced or control plants with a TYLCV C3 null mutant and evaluated the capacity of this mutant virus to establish a systemic infection. In systemic tissues of control plants, the mutant virus accumulated to very low levels, and caused only mild symptoms (Fig. 4a–c); nevertheless, upon *POLE2* silencing, both viral accumulation and symptom development were dramatically increased (Fig. 4a–c). Silencing *POLE1*, encoding the catalytic subunit of DNA polymerase ε, had a similar effect on the ability of a TYLCV C3 null mutant to establish a systemic infection (Fig. 4d). The finding that silencing subunits of DNA polymerase ε can complement the lack of C3 strengthens the idea that one of the functions of this viral protein is to counter the negative effect of DNA polymerase ε on viral replication.

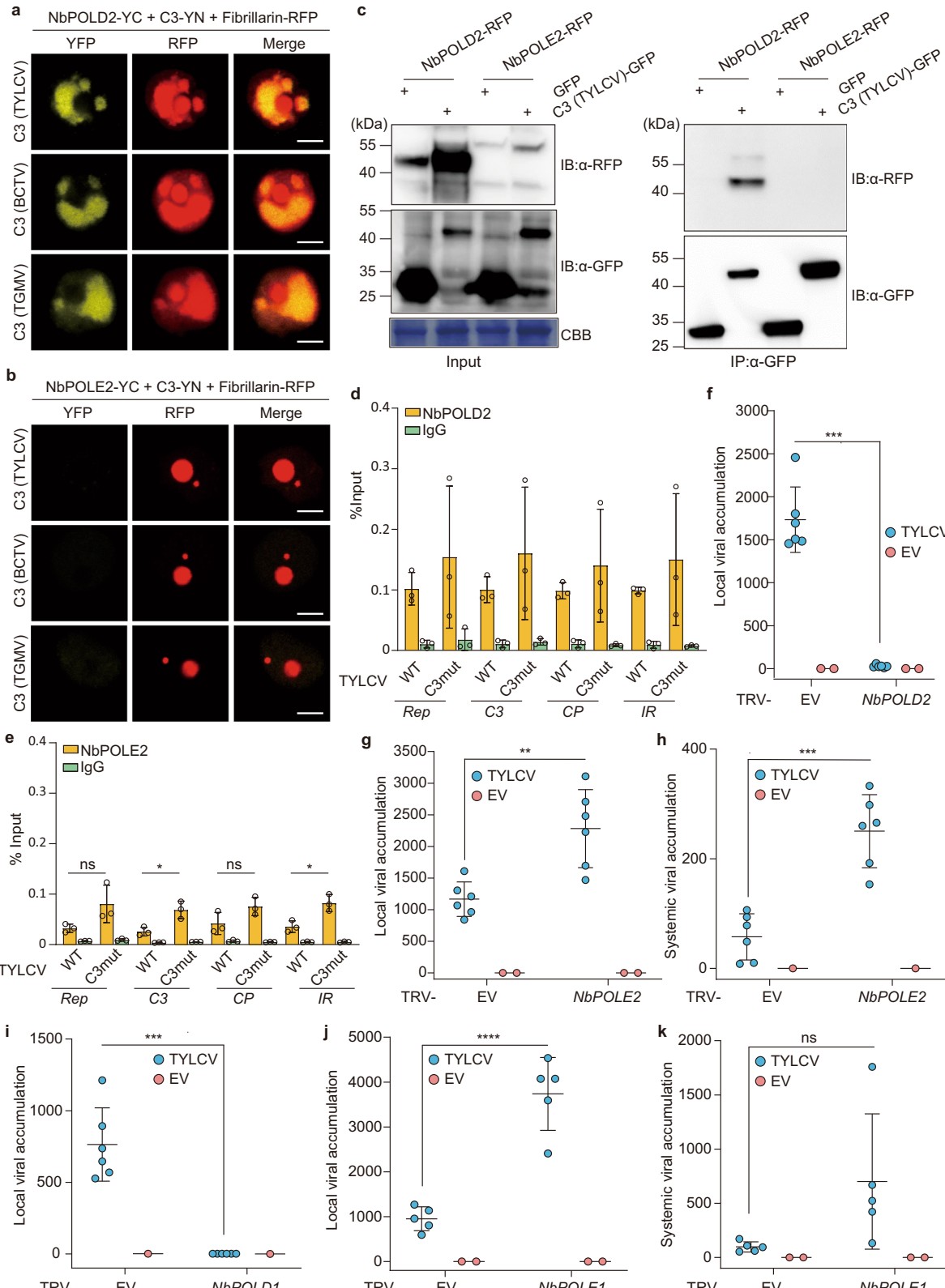

## Discussion

Taken together, our results identify DNA polymerases α and δ as required for replication of geminiviruses in their host plants. The role of replicative DNA polymerases in this process is in agreement with the previous observation that treatment with aphidicolin, an inhibitor of DNA polymerases α, δ, and ε, impairs geminiviral accumulation in plants[28]. DNA polymerase α, but not

DNA polymerase δ, is essential for the accumulation of the dsDNA replicative intermediate (Fig. 3b). Despite its basal low processivity, it remains to be determined whether DNA polymerase α performs the synthesis of the viral CS alone, or it does so in conjunction with a yet-to-be-identified polymerase. DNA polymerase δ is required for the subsequent synthesis of new viral ssDNA; a contribution of DNA polymerase α, or of additional

**Fig. 2 DNA polymerase δ, but not DNA polymerase ε, interacts with the geminivirus-encoded C3 protein and is required for geminiviral replication. a, b** NbPOLD2 (**a**), but not NbPOLE2 (**b**), interacts with C3 from TYLCV, BCTV, and TGMV in BiFC assays upon transient expression in *N. benthamiana*. Fibrillarin-RFP marks the nucleolus and the Cajal body. Scale bar: 5 μm. Negative controls are shown in Supplementary Fig. 1. **c** C3-GFP co-immunoprecipitates NbPOLD2-RFP (left), but not NbPOLE2-RFP (right), upon transient expression in *N. benthamiana*. IP: immunoprecipitate; IB: immunoblotting; CBB: Coomassie brilliant blue. The predicted protein sizes are as follows: NbPOLD2-RFP, ~46.5 kDa; NbPOLE2-RFP, ~63 kDa; C3 (TYLCV)-GFP, ~42 kDa; GFP, ~26 kDa. Full blots and membranes can be found in the Source data file. **d, e** NbPOLD2 (**d**) and NbPOLE2 (**e**) bind the TYLCV genome in ChIP assays. The location of the amplified sequences at different genomic regions is shown in Supplementary Fig. 3a; results for additional genomic regions are shown in Supplementary Fig. 7. Data are the mean of three independent biological replicates; error bars indicate SD. Asterisks indicate a statistically significant difference according to a two-sided Student's *t* test (*$P < 0.05$). ns: not significant. **f, g** Viral accumulation in local TYLCV infections (3 days post inoculation) in *POLD2*-silenced (TRV-NbPOLD2) (**f**), *POLE2*-silenced (TRV-NbPOLE2) (**g**), or control (TRV) *N. benthamiana* plants measured by qPCR. Plants inoculated with the empty vector (EV) are used as negative control. Data are the mean of six independent biological replicates; error bars represent SD. The phenotype of silenced plants and silencing efficiency are presented in Supplementary Fig. 4. **h** Viral accumulation in systemic TYLCV infections (2 weeks post inoculation) in *POLE2*-silenced (TRV-NbPOLE2) or control (TRV) *N. benthamiana* plants measured by qPCR. Plants inoculated with the empty vector (EV) are used as negative control. Data are the mean of six independent biological replicates; error bars represent SD. The 25S ribosomal DNA interspacer (*ITS*) was used as a reference gene; values are presented relative to *ITS*. **i, j** Viral accumulation in local TYLCV infections (3 days post inoculation) in *POLD1*-silenced (TRV-NbPOLD1) (**i**), *POLE1*-silenced (TRV-NbPOLE1) (**j**), or control (TRV) *N. benthamiana* plants measured by qPCR. Plants inoculated with the empty vector (EV) are used as negative control. Data are the mean of six (**i**) or five (**j**) independent biological replicates; error bars represent SD. The phenotype of silenced plants and silencing efficiency are presented in Supplementary Fig. 4. **k** Viral accumulation in systemic TYLCV infections (2 weeks post inoculation) in *POLE1*-silenced (TRV-NbPOLE1) or control (TRV) *N. benthamiana* plants measured by qPCR. Plants inoculated with the empty vector (EV) are used as negative control. Data are the mean of five independent biological replicates; error bars represent SD. The 25S ribosomal DNA interspacer (*ITS*) was used as a reference gene; values are presented relative to *ITS*. All experiments were repeated at least three times with similar results, with the exception of the ChIP assays, which were repeated twice. Two-sided Student's *t* test (**f, g, h, i, j, k**) was performed to test statistical significance (****$P < 0.0001$; ***$P < 0.001$; **$P < 0.01$). ns: not significant. The original data from all experiments and replicates can be found in the Source data file.

DNA polymerases, to this step of the viral cycle cannot be ruled out at this point. Interestingly, TYLCV has been recently proven to replicate in the insect vector in a DNA polymerase δ-dependent manner[29], which raises the idea that the mechanisms replicating geminiviruses might be conserved between the animal and plant kingdoms.

Protein–protein interactions and functional data suggest a model in which the geminivirus-encoded replication enhancer C3 acts selectively recruiting DNA polymerase δ over the non-productive DNA polymerase ε (Fig. 4e); this function of C3 would explain the long-standing observation that the lack of this viral protein decreases the accumulation of viral DNA[6,8,9,12,13,30]. Notably, a C3 null mutant TYLCV shows a decrease in both VS and CS (Fig. 3b–e), in agreement with previous observations[9]; this raises the possibility that C3 also plays a role in the initial productive recruitment of DNA polymerase α, an idea supported by its physical interaction with POLA2 (Fig. 1). Nevertheless, it should be noted that not all geminiviruses are described to encode a C3 protein, and hence alternative mechanisms for the selective recruitment of DNA polymerases might be in place. It also needs to be considered that viral replication-related proteins are not present during the first phase of infection; hence, the initial recruitment of DNA polymerase α and any other factors required for the synthesis of the dsDNA must happen without the contribution of these viral effectors.

In yeast, DNA polymerase δ acts replicating the leading strand during double-strand break repair, a process in which it is error-prone[31,32]; during the replication of the geminiviral genome, a similar decrease in fidelity might explain the high mutation rate of these viruses. DNA polymerase α, which lacks proofreading activity, may also introduce errors during the synthesis of the dsDNA intermediate. Alternatively or additionally, the accumulation of mutations in geminiviral genomes might not be linked to the replicative process, but occur in the ssDNA form as a result of oxidative damage (reviewed in ref. [33]).

Geminiviruses belong to the Eukaryotic Circular Rep-Encoding Single-Stranded DNA (CRESS DNA) viruses, a virus phylum that encompasses Rep-encoding ssDNA viruses with a likely common ancestor infecting organisms from different kingdoms of life, including animals, plants, and fungi[33,34]; since CRESS DNA viruses are expected to display similar strategies for the replication of their genomes, the identification of the DNA polymerases mediating replication of geminiviruses could have an impact on host–virus interactions beyond those involving this viral family.

## Methods

**Plant materials.** *N. benthamiana* plants were grown in a controlled growth chamber under long-day conditions (LD, 16 h of light/8 h of dark) at 25 °C.

**Plasmid construction.** Plasmids and primers used for cloning are summarized in Supplementary Tables 2 and 3. The TYLCV clone used as a template is AJ489258 (GeneBank). pGTQL1211YN and pGTQL1221YC are described in[35]; vectors from the pGWB series are described in refs. [36,37]. DNA fragment cloned into the pENTR™/D-TOPO entry vector or the pDNOR-zeo entry vector (Thermo Scientific) were recombined into the corresponding destination vectors through a Gateway LR reaction (Thermo Scientific).

**Y2H assay.** Yeast two-hybrid assays were performed following the Matchmaker Yeast Two-Hybrid User Manual (Clontech).

**Identification and selection of *Nicotiana benthamiana* orthologous genes encoding DNA polymerase subunits.** Orthologs of Arabidopsis POLA1/2, POLD1/2, POLE1/2, and REV3 proteins in *N. benthamiana* were identified by BLAST (Supplementary Table 1). The proteins and coding genes used in this work were selected among the corresponding orthologs based on their gene expression in TYLCV-infected *N. benthamiana* samples ([38]; Supplementary Table 1).

**Local and systemic viral infections.** Local and systemic viral infection assays were done as described in ref. [38]. In brief, *Agrobacterium* cells carrying the TYLCV-WT, TYLCV-C3mut, or BCTV infectious clones, or an empty vector (EV) as control, were liquid-cultured in LB with appropriate antibiotics overnight. Bacterial cultures were centrifuged at $4000 \times g$ for 10 min and resuspended in infiltration buffer (10 mM MgCl$_2$, 10 mM MES, pH 5.6, and 150 μM acetosyringone). After a 4-h incubation at room temperature in the dark, bacterial cultures were used to infiltrate the underside of leaves of 4-week-old *N. benthamiana* plants (for local infection assays) or inject on the stems of 3-week-old *N. benthamiana* plants (for systemic infection assays).

**Virus-induced gene silencing (VIGS).** *Tobacco rattle virus* (TRV)-mediated virus-induced gene silencing (VIGS) assays were performed as described in[39]. TRV-NbPDS was used as a positive control[40]. Briefly, *Agrobacterium* cells carrying pTRV1- and pTRV2-based constructs were grown in LB medium overnight with appropriate antibiotics. Cultures were resuspended in the infiltration buffer

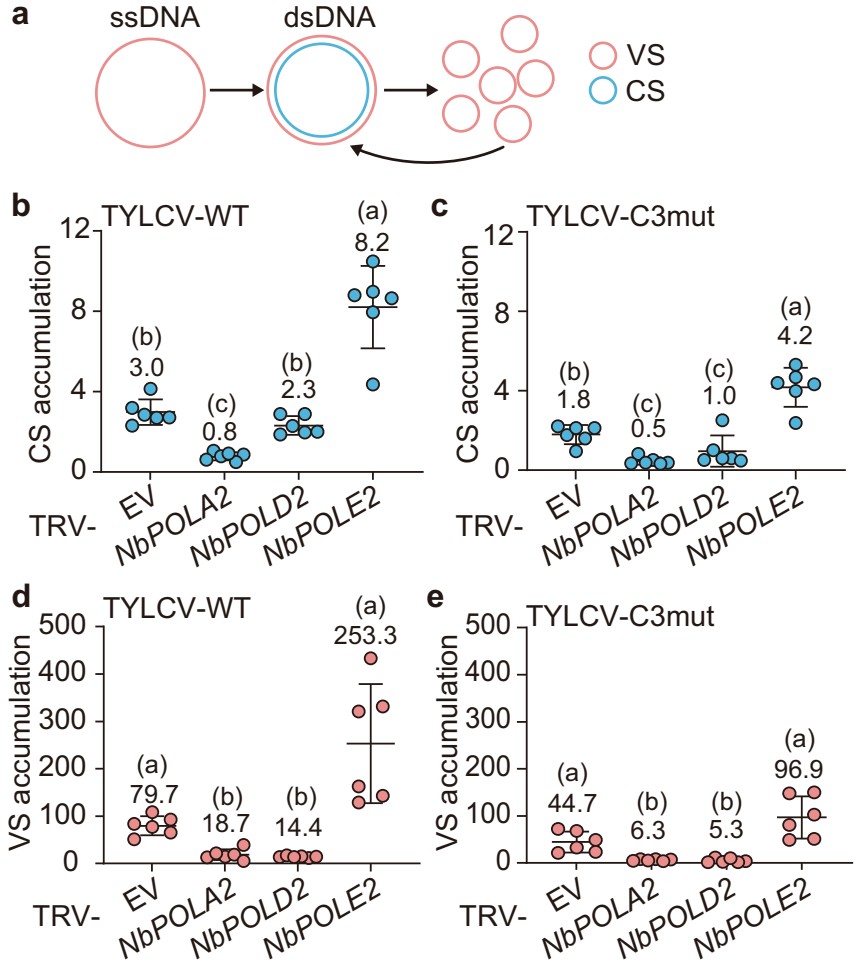

**Fig. 3 Effect of silencing *POLA2*, *POLD2*, or *POLE2* on the accumulation of viral and complementary DNA strands. a** Schematic representation of the viral DNA forms during the infection. VS: viral strand; CS: complementary strand; ssDNA: single-stranded DNA; dsDNA: double-stranded DNA. **b, c, d, e** Accumulation of complementary strand (CS) (**b, c**) and viral strand (VS) (**d, e**) during local TYLCV infections in *POLA2*-silenced (TRV-NbPOLA2), *POLD2*-silenced (TRV-NbPOLD2), *POLE2*-silenced (TRV-NbPOLE2), or empty vector control (EV) *N. benthamiana* plants, measured by qPCR at 3 days post inoculation. Data are the mean of six independent biological replicates; error bars represent SD. The 25S ribosomal DNA interspacer (*ITS*) was used as a reference gene; values are presented relative to *ITS*. TYLCV-WT: wild-type TYLCV; TYLCV-C3mut: C3 null TYLCV mutant. Mean values are shown. Letters indicate a statistically significant difference according to one-way ANOVA-Welch (in **b**: degrees of freedom df = 3, F value = 50.98; in **d**: degrees of freedom df = 3, F value = 18.42; in **e**: degrees of freedom df = 3, F value = 17.70) followed by Games–Howell's multiple comparison test ($P < 0.05$), or according to one-way ANOVA (in **c**: degrees of freedom df = 3, F value = 32.27) followed by Tukey's multiple comparison test ($P < 0.05$). The original data from all experiments and replicates can be found in the Source data file.

(10 mM MgCl$_2$, 10 mM MES, pH 5.6, and 150 μM acetosyringone) and incubated at room temperature for 4 h in the dark. Mixed cell cultures were used to inoculate 2-week-old *N. benthamiana* plants. Two weeks later, plants were used for local infection assays. For systemic infection assays, *Agrobacterium* cells carrying the virus infectious clones were co-infiltrated with pTRV1- and pTRV2-based constructs.

**Quantitative real-time PCR (qPCR) and reverse transcription PCR (RT-qPCR).** To determine viral accumulation, total DNA was extracted from *N. benthamiana* leaves (from infiltrated leaves in local infection assays and from apical leaves in systemic infection assays) using the CTAB method[41]. Quantitative real-time PCR (qPCR) was performed with primers to amplify *Rep* (Supplementary Table 3). The 25S ribosomal DNA interspacer (*ITS*) was used as a reference gene (Supplementary Table 3).

The quantification of viral and complementary strand in local infections was performed following ref. [27].

To detect gene expression in *N. benthamiana*, total RNA was extracted from leaves by using Plant RNA kit (OMEGA Bio-tek). cDNA was synthesized using the iScript™ gDNA clear cDNA Synthesis Kit (Bio-Rad) according to the manufacturer's instructions. *NbActin* was used as a reference gene. qPCR and RT-qPCR were performed in a Bio-Rad CFX96 real-time system with Hieff™ qPCR

SYBR Green Master Mix (Yeason). The reactions were done as follows: 3 min at 95 °C, 40 cycles consisting of 15 s at 95 °C, and 30 s at 60 °C. Primers used are described in Supplementary Table 3.

**Chromatin immunoprecipitation (ChIP) assay.** ChIP assay was performed as described previously[42]. *Agrobacterium* clones carrying the binary vectors to express *NbPOLA2-*, *NbPOLD2-*, or *NbPOLE2-GFP* were co-infiltrated with those carrying the TYLCV or TYLCV-C3mut infectious clones in *N. benthamiana* leaves. The infiltrated tissue was collected and cross-linked with 1% formaldehyde in 1xPBS buffer at 2 dpi; 3 μg of antibody were used. ChIP products were diluted into 200 μL of ddH$_2$O and inputs were diluted at a 1:100 ratio, and analyzed by qPCR. Anti-GFP antibody (Abcam, ab290) and IgG (Sigma, I5006) were used in this assay. The primers used in this experiment are listed in Supplementary Table 3.

**Protein extraction and co-immunoprecipitation (co-IP) assays.** *Agrobacterium* cells carrying the appropriate constructs were infiltrated in *N. benthamiana* leaves and collected at 2 dpi. After grinding the agroinfiltrated tissue in liquid nitrogen, nuclei were extracted as in ChIP assay, and then subjected to protein extraction and co-immunoprecipitation assays with GFP-Trap beads (Chromotek, Germany; Smart Lifesciences, SA070005) as described in ref. [19]. The antibodies used are as follows: anti-GFP (Abiocode, M0802-3a), anti-RFP (Chromotek, 5F8), anti-mouse

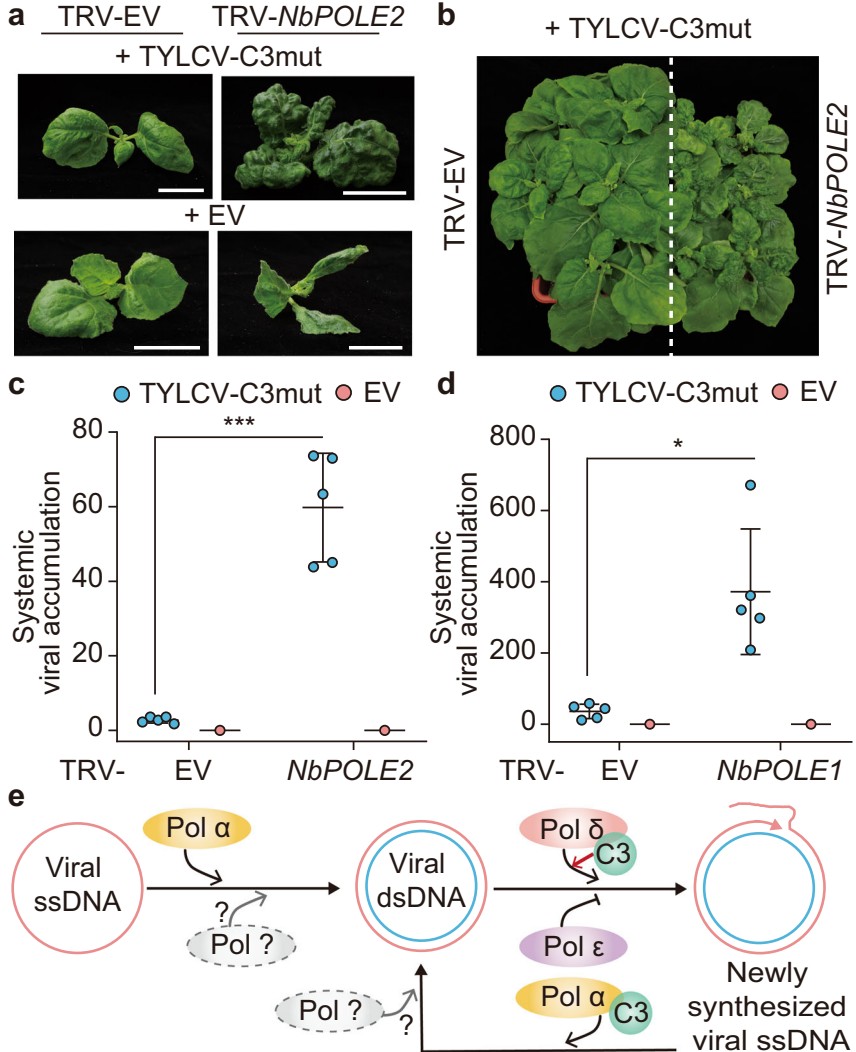

**Fig. 4 Silencing subunits of DNA polymerase ε enables systemic infection of a C3 null mutant geminivirus. a, b** Symptoms of *POLE2*-silenced (TRV-NbPOLE2, right) or control (TRV-EV, left) *N. benthamiana* plants inoculated with a C3 null mutant TYLCV at 4 weeks post inoculation. EV: empty vector control. Scale bar: 2 cm. **c, d** Viral accumulation in systemic TYLCV infections (4 weeks post inoculation) in *POLE2*-silenced (TRV-NbPOLE2) (**c**), *POLE1*-silenced (TRV-NbPOLE1) (**d**), or control (TRV-EV) *N. benthamiana* plants measured by qPCR. Plants inoculated with the empty vector (EV) are used as negative control. Data are the mean of five independent biological replicates; error bars represent SD. The 25S ribosomal DNA interspacer (*ITS*) was used as a reference gene; values are presented relative to *ITS*. Asterisks indicate a statistically significant difference according to a two-sided Student's *t* test (***$P < 0.001$; *$P < 0.05$). **e** Hypothetical model of the role of DNA polymerases α and δ in the replication of the geminiviral genome. DNA polymerase α is required to convert the viral ssDNA genome to the dsDNA replicative intermediate, which is then replicated by DNA polymerase δ to produce new viral ssDNA. The virus-encoded C3 protein interacts with DNA polymerase α (POLA2) and DNA polymerase δ (POLD2), and selectively recruits the latter over the non-productive DNA polymerase ε. The original data from all experiments and replicates can be found in the Source data file.

IgG (Sigma, A2554), and anti-rat IgG (Abcam, ab7097). Primary antibodies were diluted 1:5000; secondary antibodies were diluted 1:15,000.

**Protein subcellular localization**. For subcellular localization, GFP- or RFP-tagged proteins were transiently expressed in *N. benthamiana* leaves and imaged with a Leica TCS SMD confocal microscope using the preset settings for GFP (Ex: 488 nm, Em: 500–550 nm) and RFP (Ex: 554 nm, Em: 570–620 nm).

**Bimolecular fluorescence complementation (BiFC)**. Bimolecular fluorescence complementation (BiFC) assays were performed in *N. benthamiana* leaves as described in ref. [35]. *Agrobacterium* cells carrying the appropriate BiFC clones and RFP-Fibrillarin[43] were infiltrated on 4-week-old *N. benthamiana* plants with 1-mL needleless syringe. Imaging was performed 2 days later under a Leica TCS SMD confocal microscope by using the preset sequential scan settings for YFP (Ex: 514 nm, Em: 525–575 nm) and for RFP (Ex: 554 nm, Em: 570–620 nm).

**Reporting summary**. Further information on experimental design is available in the Nature Research Reporting Summary linked to this paper.

## Data availability

All data generated or analyzed during this study are included in this published article (and its supplementary information files). Source data are provided with this paper.

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

## Acknowledgements
The authors thank all members of Rosa Lozano-Duran's lab and Alberto Macho's lab for fruitful discussions; Xinyu Jian, Aurora Luque, and the PSC Cell Biology Facility for technical assistance; Emmanuel Aguilar for his invaluable help with statistical analyses; and Alberto P Macho, Chaonan Shi, and Laura Medina-Puche for critical reading of the paper. This work was supported by the Strategic Priority Research Program of the Chinese Academy of Sciences (Grant No. XDB27040206) and the Shanghai Center for Plant Stress Biology from the Chinese Academy of Sciences.

## Author contributions
Conceptualization: R.L.-D. and E.R.B.; investigation: M.W., H.W., H.T., and S.P.; writing: M.W. and R.L.-D.; visualization: H.T., M.W., and H.W.; supervision: R.L.-D.; funding acquisition: R.L.-D. and Q.L.

## Competing interests
The authors declare no competing interests.
