## [Peer Review File · Nature Communications]

We would like to thank the reviewers for their careful assessment of our manuscript and their suggestions, which we believe have contributed to improving both content and form. Following the reviewers' advice, we have now made a number of changes to our manuscript, including the addition of new experimental data further supporting our conclusions; a point-by-point response to the reviewers' comments can be found below. (Please note that line numbers, where indicated, refer to the clean version of the manuscript).

REVIEWER COMMENTS

Reviewer #1 (Remarks to the Author):

The manuscript describes experiments examining the function of the C3 protein during geminivirus replication. They provide evidence that C3 interacts with regulatory subunits of DNA POLA and POLD – two of the three replicative DNA polymerases in eukaryotes. VIGS of POLA2 and POLD2 reduced viral DNA levels, while VIGS of POLE2 increased viral DNA levels. They also show that POLA2, POLD2 and POLE2 interact with viral DNA, and propose that C3 interactions with POLA2 and POLD2 facilitate their recruitment to viral DNA and relieve POLE2 inhibition of viral replication. Based on quantification of viral virion sense (VS) and complementary sense (CS) strands, the authors propose that POLA mediates CS synthesis, while POLD is primarily responsible for VS synthesis. An overarching assumption in the manuscript is that POLA, POLD and POLE activities can be assessed by studying POLA2, POLD2 and POLE2. There are no studies directly examining the catalytic subunits of the DNA polymerases. The authors need to provide a strong rationale (and references) supporting their assumption that the regulatory subunits are valid proxies for DNA polymerase activity in a viral system.

- In order to address this point, we have added the silencing of the genes encoding the catalytic subunits of DNA polymerases α , δ , and ϵ (POLA1, POLD1, and POLE1, respectively) (new Figure 1i, Figure 2i-k, Figure 4d; the phenotype of the silenced plants and the silencing efficiency are presented in Supplementary figure 4). The results obtained when silencing these subunits are similar to the ones previously obtained when silencing the regulatory subunits (POLA2, POLD2, and POLE2), supporting the idea that it is indeed the activity of the DNA polymerase α/δ holocomplex, and not of their regulatory subunits specifically, what is required for geminiviral replication. Since in yeast and mammalian cells the translesion DNA polymerase ζ shares two subunits with DNA polymerase delta, namely POLD2 and POLD3 (Baranovski et al., 2012, *JBC*; Johnson et al., 2012, *PNAS*), we have also added a new experiment in which we test the effect of silencing the gene encoding the DNA polymerase ζ -specific catalytic subunit, REV3, on TYLCV accumulation. As shown in new Figure S9, *REV3* silencing does not affect the accumulation of viral DNA, ruling out a requirement of this polymerase for viral replication.

There are also aspects of the interpretation of the results that need to be addressed.

- The authors do not perform statistical analyses of their viral DNA accumulation results. Given the large variation between replicates for some treatments, this is a problem. In Fig. 3C, VS levels (not just CS levels) appear to be down in the POLA2 VIGS treatment. In addition, it is not clear if POLE2 VIGS differs from the controls in Fig. 3B. It is recommended that the authors determine if there are Statistically significant differences in the levels of total viral DNA, CS, and VS in the comparisons throughout the manuscript.

- **Following the reviewer's advice, statistical analysis has now been added to the viral DNA accumulation experiments (Figure 1g-l; Figure 2f-k; Figure 3b,c; Figure 4c,d; Supplementary figure 6; Supplementary figure 9). In Figure 3b and c, for clarity, we have also added the mean values to the graph. Indeed, VS levels are also diminished in POLA2-silenced plants. Since POLA2 seems to be required for the CS synthesis and the dsDNA acts as template for the subsequent RCR, we speculate that the impact on VS is the effect of reduced template availability; nevertheless, a contribution of DNA polymerase α to RCR cannot be ruled out. This is now specifically mentioned in the text (Lines 174-175 and 204-207).**

- The authors do not address the observation that POLD2 VIGS results in a large decrease in POLA2 as well as POLD2 transcript levels. (The opposite is not true for POLD2 transcripts during POLA2 silencing.) This raises the possibility that POLA2 transcript reductions mediate the decrease in VS levels, not lower POLD2 transcripts, calling a main conclusion of the manuscript into question. The possibility that POLA is the primary DNA polymerase for both CS and VS synthesis is also consistent with CS accumulation depending on the generation of new VS DNA to serve as template for CS synthesis. Importantly, VS levels are also reduced when POLA2 is silenced.

- **Although silencing POLD2 reduces the accumulation of POLA2 transcripts (Supplementary figure 8b), it should be noted that i) CS levels are not affected in POLD2-silenced plants (Figure 3b), in contrast to POLA2-silenced plants; and ii) a similar reduction in POLA2 transcripts is observed upon POLE2 silencing (Supplementary figure 8c), which nevertheless has the opposite effect on viral replication. However, in order to fully assess the possibility suggested by the reviewer, we have now measured the accumulation of the transcripts encoding the different subunits of DNA polymerase α in POLD1- and POLD2-silenced plants. As shown in Supplementary figure 8d, silencing POLD1, as opposed to silencing POLD2 (Supplementary figure 8e), does not negatively affect accumulation of these transcripts. Since POLD1 silencing impairs viral DNA accumulation similarly to POLD2 silencing, we strongly believe this effect depends on the activity of the DNA polymerase δ holoenzyme.**

The following experimental points also need to be addressed.

- The immunoprecipitation results for C3 and POLE2 (Fig. 2d) are not convincing. The blot is very dark for POLE2, and there appears to be a band in the GSP-C3 lane that is different from a background band in the control lane. The BiFC results support the authors' conclusion but do

not rule out that NbPOLE2-YC is not expressed. The authors need to strengthen their results that C3 and POLE2 do not interact.

- **Following the reviewer's advice, we have now repeated the co-IP experiment in Figure 2c and present a clearer blot. Additionally, we have confirmed expression of NbPOLE2-YC by western blot (new Supplementary figure 1b).**

- It is not clear what part of the gel is shown in the CBB panels of the IP experiments.

- **For the CBB panel, we chose a clear ~55 kDa band in all gels, possibly corresponding to the large subunit of Rubisco. Images of the full membranes are now available in the Source data file.**

- No controls are included in the BiFC assays to validate specificity of interaction. It is standard procedure to analyze noninteracting fusions in parallel. The negative controls should be shown in supplementary results.

- **We have now included two nuclear-localized proteins as negative controls, IMPA2 from *Arabidopsis thaliana* and V2 from TYLCV, in Supplementary figure 1. Expression of all fusion proteins is confirmed in Supplementary figure 1b.**

- The controls in Fig. 1B needs to be described.

- **We thank the reviewer for pointing out this omission; this information has now been added to the corresponding figure legend.**

The experiments are novel and could be a major breakthrough in our understanding of how C3 enhances begomovirus and curtovirus replication. The experiments may also provide insight into how other eukaryotic CRESS viruses replicate their genomes and, as such, would be of interest to the virology community. However, it is important to note that not all geminiviruses and other CRESS viruses encode C3 proteins, and it is unclear how widely applicable the results will be.

- **We absolutely agree with the reviewer. We have now added a cautionary note indicating that not all geminiviruses encode C3 proteins, and therefore alternative mechanisms for the selective recruitment of DNA polymerase δ over ϵ might exist (Line 217-220). We believe, however, that the identification of DNA polymerases α and δ as required for the replication of the geminiviral DNA might be relevant for other CRESS viruses, which likely have conserved replication mechanisms; this is now specifically mentioned in Line 234-237.**

Reviewer #2 (Remarks to the Author):

The manuscript from Wu et al describes work on the role of DNA polymerases in replication of geminivirus genomic DNA. Specifically, the authors show that DNA polymerase α is essential to generate the viral dsDNA intermediate, but that DNA polymerase δ mediates synthesis of the geminiviral ssDNA genome. In addition, they show that the viral C3 replication enhancer protein selectively recruits DNA polymerase δ over ϵ . The work is original and of interest to the geminivirus field but also to people in the replication field. I have some comments outlined below.

The manuscript is well written and the experiments appear to be well performed with the appropriate controls. However, it may be that the authors have written the manuscript appropriately, but I thought it lacked detailed explanation of the results with significant details concerning the experiments missing. The experiments seem to have a summary rather than describing the results in some detail for the author to follow. The details are not provided in the figure legends either. As an example, for the co-IP experiment, what are the sizes of the proteins and which ones correspond to those shown in the figure itself, what are the details of the antibodies used?

- **Following the reviewer's advice, we now provide more detailed explanations of experiments and results, and have expanded the figure legends to contain all relevant information. The predicted size of the different fusion proteins has been added to the figure legends of Figure 1c and Figure 2c; the antibodies used for immunoprecipitation and detection are indicated in the panels, and the company details are included in the relevant Methods section.**

Other comments:

On line 58 the authors indicate that the C3 proteins from BCTV and TGMV interact with NbPOLA2 (Fig. 1E). The interaction, at least by BiFC appears weaker for the Nb protein. Was this observed for co-IP or Yeast two hybrid?

- **While, as the reviewer points out, the interaction of the different C3 proteins with NbPOLA2 seems to be weaker than that with SiPOLA2 in BiFC, this difference is not observed in co-IP assays (new Figure 1c).**

What is the relevance of the speckles in Figure 1F and Supplementary figure 1

- **We thought this observation was worth reporting, because it implies a change in the subnuclear distribution of POLA2 triggered by the presence of the virus; however, the potential functional relevance of this change is at this point unknown. This has now been stated in the manuscript (Lines 84-87).**

There is no indication of how this work compares to other work previously published on C3 and replication. As an example, how does the data fit with observations that C3 mutants decrease viral DNA loads and that both ss and dsDNA levels are reduced by similar amounts? If DNA polymerase α generates the dsDNA intermediate, and DNA polymerase δ mediates synthesis of

ssDNA, and C3 selectively recruits DNA polymerase δ , then wouldn't you expect a mutation in C3 to reduce levels of ssDNA relative to dsDNA? The manuscript would benefit from a more detailed discussion.

- **Following the reviewer's advice, we have now added a specific reference to the previous results obtained with C3 mutant geminiviruses in the discussion, and elaborated on how our model could explain them (Line 201-223). Indeed, if C3 affected the recruitment of DNA polymerase δ only, then we would expect a specific effect on the accumulation of viral ssDNA, as the reviewer points out; the finding that dsDNA levels are also affected in the absence of C3 suggests that this protein might also be involved in the initial productive recruitment of DNA polymerase α , an idea supported by its physical interaction with POLA2. This possibility is included in our graphical model (Figure 4e), and is now specifically mentioned in the discussion (Line 215-217).**

Reviewer #3 (Remarks to the Author):

Geminiviruses are economically of high importance because of their impact on crop cultivation. The viral Rep protein is the only protein necessary for replication of the viral genome but is not a polymerase. The viral C3/AC3/AL3 protein, while not absolutely required for replication, enhances this process. Since geminiviruses do not encode an own polymerase they rely on host polymerases for replication of their genome. Furthermore, geminiviruses infect differentiated plant cells so the available proteome differs from that of an actively cycling cell at the moment of infection. The situation may change with time since Rep has been shown to activate DNA synthesis in differentiated cells.

The authors identified subunits of the replicative DNA polymerases α and δ in a Y2H screen with the tomato yellow leaf curl virus C3 protein. The interaction was confirmed by GFP pull-down assays and BiFC. They further show that a subunit of polymerase α , δ , and ϵ bind to the viral genome and binding is dependent on the presence of the C3 protein. Further experiments show that polymerase α and δ , but not ϵ are involved in accumulation of viral DNA in local and systemic infections.

While the finding that a geminiviral protein, C3, interacts with subunits of replicative polymerases is new, further conclusions are not well supported by the experiments and not put into context of geminivirus infection and replication. There are key questions that are not addressed or discussed. Geminiviruses need DNA polymerases for the first step of replication, complementary strand replication, after introduction of the virus by the vector into a host cell. Are polymerase α and δ present in the differentiated cells geminiviruses infect? At this time point the viral Rep protein is not yet expressed to activate the cell cycle.

- **We completely agree with the reviewer in that DNA polymerase α must be expressed prior to the viral infection in order for this model to hold. This polymerase would enable the production of the dsDNA intermediate, from which Rep would be expressed, hence the reactivation of the cell cycle could occur from that point.**
According to publicly available data, the genes encoding all four subunits of DNA polymerase α are broadly expressed in *Arabidopsis thaliana* (see Figure for Reviewers 1a; Zhang *et al.*, 2020), and are expressed in phloem companion cells, the specific cells sustaining viral replication for TYLCV and many other geminiviruses, in roots (Figure for Reviewers 1b; Brady *et al.*, 2007). Similarly, we can also detect the expression of at least one orthologue per subunit in *Nicotiana benthamiana* leaves by RNA-seq (Supplementary table 1; Wu *et al.*, 2019). Therefore, DNA polymerase α seems to be expressed in all plant tissues tested, at least in *A. thaliana*, making our proposed model feasible. Importantly, this model is supported by functional data: silencing genes encoding subunits of the DNA polymerase α holocomplex, namely POLA1 and POLA2, impairs the accumulation of geminiviral DNA in both its dsDNA and ssDNA forms (Figure 1g-i; Figure 3b, c).

Figure for Reviewers 1. Expression of genes encoding the subunits of DNA polymerase α in different tissues (a; Zhang *et al.* 2020) and in roots, including phloem companion cells (b; Brady *et al.* 2007), in *A. thaliana*. Images in b are taken from eFP browser (bar.utoronto.ca/efp2/Arabidopsis/Arabidopsis_eFPBrowser2.html).

REFERENCES

Brady, S.M., Orlando, D.A., Lee, J.-Y., Wang, J.Y., Koch, J., Dinneny, J.R., Mace, D., Ohler, U., Benfey, P.N., 2007. A high-resolution root spatiotemporal map reveals dominant expression patterns. *Science* 318, 801-806.

Zhang, H., Zhang, F., Yu, Y., Feng, L., Jia, J., Liu, B., Li, B., Guo, H., Zhai, J., 2020. A Comprehensive Online Database for Exploring approximately 20,000 Public Arabidopsis RNA-Seq Libraries. *Mol Plant* 13, 1231-1233.

The manuscript text suggests that DNA polymerase α and δ are solely responsible for replication of the viral genome although the role of translesion DNA polymerases in geminivirus replication has been shown recently. It is known that translesion DNA polymerases are constitutively expressed at high levels in differentiated plant tissues (at least in arabidopsis).

- Although a contribution of translesion DNA polymerases to geminiviral replication has been recently shown, these polymerases were not required for geminivirus replication (Richter *et al.*, 2016); this reference has now been included in the text (Line 55-57). Additionally, and since DNA polymerase δ shares two subunits, namely POLD2 and POLD3, with the translesion DNA polymerase ζ , we have also tested the requirement of this polymerase for TYLCV replication by silencing the gene encoding the catalytic subunit of this holocomplex, REV3. Our results show that REV3 is not required for viral accumulation (Supplementary figure 9). Nevertheless, a role of additional DNA polymerases in geminivirus replication cannot be ruled out at this point, hence we have modified the text to make this clear (Lines 203-207).

Specific comments:

- Line 32: there are newer references that are more appropriate to cite here.

> We have now added Rojas *et al.*, 2018, to this statement.

- Lines 36/37: multiple copies

> We thank the reviewer for pointing out this mistake, which has now been corrected.

- Lines 49/50: But it is known that translesion DNA polymerases play a role in geminivirus replication.

- The text has now been modified.

- Line 61: Instead of including the whole virus family it would be more appropriate to state that the interaction is conserved at least in these two genera.

- We have modified the text accordingly.

- Lines 64-70: Very long sentence – would be easier to read if broken down into two or three sentences.

- We have now divided this sentence in two.

- Lines 91-93: Is there any explanation for the observation?

- We do not have an explanation for this observation at the moment.

- Line 102: Here again, it would be more appropriate to restrict the statement to the two virus genera of which examples have been tested.

- We have modified the sentence accordingly.

- Lines 134/135: The statement seems a bit too strong and should be down-weighted. Silencing of one polymerase alone might disturb the well-balanced levels of polymerases making it difficult to make conclusions.

- We agree with the reviewer in that this is not unequivocal proof, hence we mention that this is one possibility and that the experimental results support this idea.

- Lines 149/150: Here again, the manuscript would benefit of including translesion DNA polymerases in the discussion.

- Here, we are restricting the discussion to the DNA polymerases shown to be required for viral DNA replication; hence, we have left translesion DNA polymerases out, although they are specifically mentioned at other points in the text.

- Lines 151-155: This sentence seems to be a bit far-fetched.

- CRESS DNA viruses share a Rep protein, ssDNA genomes, and rolling-circle replication; therefore, it is expected that they will likely share molecular mechanisms underlying the replication of their genomes. In this context, we think it is possible that the identification of the processive DNA polymerases α and δ as required for the replication of geminiviruses might be extensible to other groups in this phylum, something that we think worth mentioning.

- Line 181: There is no description of local and systemic viral infection in reference 30?

- We thank the reviewer for pointing out this mistake; the correct reference is Wu *et al.*, 2019.

- Line 192: Virus-induced gene silencing is also not described in reference 30?

- TRV-mediated virus-induced gene silencing is described in reference 30.

- Line 218: ChIP assay refers to reference 32 that refers to yet another paper... Nevertheless, the description of the ChIP assay is very long – what are the modifications to the referred papers?

- We have now shortened this section to include only the reference and the details specific for our experimental conditions.

- Lines 403/404: Controls are not described.

- The description of the controls has now been added.

- Lines 404-406: What is shown in the cut-outs of the Coomassie-stained gels (no marker, no explanation)? What are the expected/known molecular weights of the proteins in the co-immunoprecipitation assays? In Figure 1d the IB anti-RFP shows a much lower amount of NbPOLA2-RFP for the GFP control compared to C3-GFP. This would require a much longer exposure of the IB after IP to see if a faint band is present in the control.

- We have now repeated the blots (new Figure 1c and Figure 2c) and added the predicted protein sizes to the figure legends. For the CBB panel, we chose a clear ~55 kDa band in all gels, possibly corresponding to the large subunit of Rubisco. Images of the full membranes are now available in the Source data file.

- Lines 427-429: Some gels and blots are so dark that bands are not or almost not visible. Again, it is not explained what the Coomassie-stained band is showing.

- We now show new, clearer blots (new Figure 1c and Figure 2c).

- Lines 429-432: Why are the error bars for C3mut so large? It would be helpful if the results of the ChIP assays would be shown with the same scale to compare it for the different polymerases.

- We do not have an explanation for this observation at the moment, but it is highly reproducible. Following the reviewer's advice, all ChIP results are now shown with the same scale.

- Lines 432-436: The assay seems to have more variability as each single plot suggests. If one compares the local viral accumulation after TYLCV infections in the TRV-EV background in Fig. 1h, Fig. 2g and Fig. 2h one would expect similar results which is not the case. In this context, it would be again helpful if the plots would have the same scales (one for local viral accumulation, one for systemic viral accumulation).

- Each biological replicate is independent, and therefore presents intrinsic variation. Nevertheless, each replicate includes its own negative control (TRV-EV), which is what all other values are compared to. The relative changes in all

different biological replicates are highly reproducible. The original values of each individual replicate are now available in the Source Data file.

In general, the references used should be revised thoroughly.

- We have now added a number of new references to the manuscript.

REVIEWER COMMENTS

Reviewer #1 (Remarks to the Author):

The manuscript describes experiments examining the function of the C3 protein during geminivirus replication. They provide evidence that C3 interacts with regulatory subunits of DNA POLA and POLD – two of the three replicative DNA polymerases in eukaryotes. VIGS of POLA2 and POLD2 reduced viral DNA levels, while VIGS of POLE2 increased viral DNA levels. They also show that POLA2, POLD2 and POLE2 interact with viral DNA, and propose that C3 interactions with POLA2 and POLD2 facilitate their recruitment to viral DNA and relieve POLE2 inhibition of viral replication. Based on quantification of viral virion sense (VS) and complementary sense (CS) strands, the authors propose that POLA mediates CS synthesis, while POLD is primarily responsible for VS synthesis. An overarching assumption in the manuscript is that POLA, POLD and POLE activities can be assessed by studying POLA2, POLD2 and POLE2. There are no studies directly examining the catalytic subunits of the DNA polymerases. The authors need to provide a strong rationale (and references) supporting their assumption that the regulatory subunits are valid proxies for DNA polymerase activity in a viral system.

There are also aspects of the interpretation of the results that need to be addressed.

- The authors do not perform statistical analyses of their viral DNA accumulation results. Given the large variation between replicates for some treatments, this is a problem. In Fig. 3C, VS levels (not just CS levels) appear to be down in the POLA2 VIGS treatment. In addition, it is not clear if POLE2 VIGS differs from the controls in Fig. 3B. It is recommended that the authors determine if there are Statistically significant differences in the levels of total viral DNA, CS, and VS in the comparisons throughout the manuscript.
- The authors do not address the observation that POLD2 VIGS results in a large decrease in POLA2 as well as POLD2 transcript levels. (The opposite is not true for POLD2 transcripts during POLA2 silencing.) This raises the possibility that POLA2 transcript reductions mediate the decrease in VS levels, not lower POLD2 transcripts, calling a main conclusion of the manuscript into question. The possibility that POLA is the primary DNA polymerase for both CS and VS synthesis is also consistent with CS accumulation depending on the generation of new VS DNA to serve as template for CS synthesis. Importantly, VS levels are also reduced when POLA2 is silenced.

The following experimental points also need to be addressed.

- The immunoprecipitation results for C3 and POLE2 (Fig. 2d) are not convincing. The blot is very dark for POLE2, and there appears to be a band in the GSP-C3 lane that is different from a background band in the control lane. The BiFC results support the authors' conclusion but do not rule out that NbPOLE2-YC is not expressed. The authors need to strengthen their results that C3 and POLE2 do not interact.
- It is not clear what part of the gel is shown in the CBB panels of the IP experiments.
- No controls are included in the BiFC assays to validate specificity of interaction. It is standard procedure to analyze noninteracting fusions in parallel. The negative controls should be shown in supplementary results.
- The controls in Fig. 1B needs to be described.

The experiments are novel and could be a major breakthrough in our understanding of how C3 enhances begomovirus and curtovirus replication. The experiments may also provide insight into how other eukaryotic CRESS viruses replicate their genomes and, as such, would be of interest to the virology community. However, it is important to note that not all geminiviruses and other CRESS viruses encode C3 proteins, and it is unclear how widely applicable the results will be.

Linda Hanley-Bowdoin

Reviewer #2 (Remarks to the Author):

The manuscript from Wu et al describes work on the role of DNA polymerases in replication of geminivirus genomic DNA. Specifically, the authors show that DNA polymerase α is essential to generate the viral dsDNA intermediate, but that DNA polymerase δ mediates synthesis of the geminiviral ssDNA genome. In addition, they show that the viral C3 replication enhancer protein selectively recruits DNA polymerase δ over ϵ . The work is original and of interest to the geminivirus field but also to people in the replication field. I have some comments outlined below.

The manuscript is well written and the experiments appear to be well performed with the appropriate controls. However, it may be that the authors have written the manuscript appropriately, but I thought it lacked detailed explanation of the results with significant details concerning the experiments missing. The experiments seem to have a summary rather than describing the results in some detail for the author to follow. The details are not provided in the figure legends either. As an example, for the co-IP experiment, what are the sizes of the proteins and which ones correspond to those shown in the figure itself, what are the details of the antibodies used?

Other comments:

On line 58 the authors indicate that the C3 proteins from BCTV and TGMV interact with NbPOLA2 (Fig. 1E). The interaction, at least by BiFC appears weaker for the Nb protein. Was this observed for co-IP or Yeast two hybrid?

What is the relevance of the speckles in Figure 1F and Supplementary figure 1

There is no indication of how this work compares to other work previously published on C3 and replication. As an example, how does the data fit with observations that C3 mutants decrease viral DNA loads and that both ss and dsDNA levels are reduced by similar amounts? If DNA polymerase α generates the dsDNA intermediate, and DNA polymerase δ mediates synthesis of ssDNA, and C3 selectively recruits DNA polymerase δ , then wouldn't you expect a mutation in C3 to reduce levels of ssDNA relative to dsDNA? The manuscript would benefit from a more detailed discussion.

Reviewer #3 (Remarks to the Author):

Geminiviruses are economically of high importance because of their impact on crop cultivation. The viral Rep protein is the only protein necessary for replication of the viral genome but is not a polymerase. The viral C3/AC3/AL3 protein, while not absolutely required for replication, enhances this process. Since geminiviruses do not encode an own polymerase they rely on host polymerases for replication of their genome. Furthermore, geminiviruses infect differentiated plant cells so the available proteome differs from that of an actively cycling cell at the moment of infection. The situation may change with time since Rep has been shown to activate DNA synthesis in differentiated cells.

The authors identified subunits of the replicative DNA polymerases α and δ in a Y2H screen with the tomato yellow leaf curl virus C3 protein. The interaction was confirmed by GFP pull-down assays and BiFC. They further show that a subunit of polymerase α , δ , and ϵ bind to the viral genome and binding is dependent on the presence of the C3 protein. Further experiments show that polymerase α and δ , but not ϵ are involved in accumulation of viral DNA in local and systemic infections.

While the finding that a geminiviral protein, C3, interacts with subunits of replicative polymerases is new, further conclusions are not well supported by the experiments and not put into context of geminivirus infection and replication. There are key questions that are not addressed or discussed. Geminiviruses need DNA polymerases for the first step of replication, complementary strand replication, after introduction of the virus by the vector into a host cell. Are polymerase α and δ

present in the differentiated cells geminiviruses infect? At this time point the viral Rep protein is not yet expressed to activate the cell cycle. The manuscript text suggests that DNA polymerase α and δ are solely responsible for replication of the viral genome although the role of translesion DNA polymerases in geminivirus replication has been shown recently. It is known that translesion DNA polymerases are constitutively expressed at high levels in differentiated plant tissues (at least in arabidopsis).

Specific comments:

- Line 32: there are newer references that are more appropriate to cite here.
 - Lines 36/37: multiple copies
 - Lines 49/50: But it is known that translesion DNA polymerases play a role in geminivirus replication.
 - Line 61: Instead of including the whole virus family it would be more appropriate to state that the interaction is conserved at least in these two genera.
 - Lines 64-70: Very long sentence – would be easier to read if broken down into two or three sentences.
 - Lines 91-93: Is there any explanation for the observation?
 - Line 102: Here again, it would be more appropriate to restrict the statement to the two virus genera of which examples have been tested.
 - Lines 134/135: The statement seems a bit too strong and should be down-weighted. Silencing of one polymerase alone might disturb the well-balanced levels of polymerases making it difficult to make conclusions.
 - Lines 149/150: Here again, the manuscript would benefit of including translesion DNA polymerases in the discussion.
 - Lines 151-155: This sentence seems to be a bit far-fetched.
 - Line 181: There is no description of local and systemic viral infection in reference 30?
 - Line 192: Virus-induced gene silencing is also not described in reference 30?
 - Line 218: ChIP assay refers to reference 32 that refers to yet another paper... Nevertheless, the description of the ChIP assay is very long – what are the modifications to the referred papers?
 - Lines 403/404: Controls are not described.
 - Lines 404-406: What is shown in the cut-outs of the Coomassie-stained gels (no marker, no explanation)? What are the expected/known molecular weights of the proteins in the co-immunoprecipitation assays? In Figure 1d the IB anti-RFP shows a much lower amount of NbPOLA2-RFP for the GFP control compared to C3-GFP. This would require a much longer exposure of the IB after IP to see if a faint band is present in the control.
 - Lines 427-429: Some gels and blots are so dark that bands are not or almost not visible. Again, it is not explained what the Coomassie-stained band is showing.
 - Lines 429-432: Why are the error bars for C3mut so large? It would be helpful if the results of the ChIP assays would be shown with the same scale to compare it for the different polymerases.
 - Lines 432-436: The assay seems to have more variability as each single plot suggests. If one compares the local viral accumulation after TYLCV infections in the TRV-EV background in Fig. 1h, Fig. 2g and Fig. 2h one would expect similar results which is not the case. In this context, it would be again helpful if the plots would have the same scales (one for local viral accumulation, one for systemic viral accumulation).
- In general, the references used should be revised thoroughly.

REVIEWER COMMENTS

Reviewer #1 (Remarks to the Author):

I have concerns about the statistical analyses.

Why was a Mann Whitney U test used for Fig. 2k? Students' T tests were used to analyze the same types of data in Fig. 2f, g, h, i, j and in Fig. 1g, h, i.

As performed, the ANOVA analysis in Fig. 3 does not support the proposed differential roles of POLA and POLD in complementary and virion sense replication. In panel B, There are no statistical differences between complementary sense DNA levels for TYLCV-wt in the EV, POLA2, and POLD2 treatments. The same is true for TYLCV-C3. The only statistical difference is between EV (TYLCV-wt) and POLA2 (TYLCV-C3). In panel C, there were no statistical differences between between the virion sense DNA levels for the EV, POLA2, and POLD treatments for the wild-type and mutant viruses. The ANOVA analysis supports the conclusion that silencing POLE2 increase TYLCV-wt accumulation.

The authors did not provide information about their ANOVA model but it seems that they did a single analysis for each panel. If that is the case, a better approach might be to analyze the TYLCV-wt and TYLCV-C3mut results separately or include an interaction term in their model for the two viruses.

Reviewer #2 (Remarks to the Author):

This is a revised manuscript that has taken into account previous reviewer comments. My initial comments have been addressed satisfactorily and the revisions based on other reviewer comments has made the manuscript much stronger.

Reviewer #3 (Remarks to the Author):

The authors have improved the manuscript.

The authors show now that silencing of the catalytic subunits of DNA POLA, POLD and POLE, namely POLA1, POLD1 and POLE1, results in the same effect as silencing of the regulatory subunits. In both cases silencing of the respective subunits leads to a decrease of viral DNA accumulation for POLA and POLD whereas silencing of POLE subunits results in an increase of viral DNA.

The quality of the blots has been improved. Nevertheless, some aspects are still not resolved. In the co-IP experiment, the size of NbPOLE2-RFP in the input fraction doesn't match the expected size of 63 kDa, but rather seems to be in the range of 55 kDa (or even below if taking the source data file with the molecular weight marker into account). The observed lower molecular weight might be the result of a truncation of the fusion protein which might in turn influence the interaction potential of POLE2 with C3.

Expression of the proteins used in BiFCs assays is now included in the Supplementary Information. Here again, it seems there are some inconsistencies. NbPOLD2-YC has an expected size of 28 kDa which is probably not the strong band on the blot around 40 kDa. For the same reason, the band for NbPOLE2-YC (expected size of 44 kDa) is not convincing. Together with the result of the co-IP it is still unclear if POLE2 and C3 are not interacting.

The discussion has improved but still needs language polishing (e.g. lines 207-209, 226/227).

Specific comments:

- Line 94: What is meant with thicker leaves? Is the whole leave (leave blade) thicker or the veins? It is not clear from the images in Supplementary Figure 4.

- I still didn't find a description for TRV-mediated gene-silencing in reference 39 (Medina-Puche et al. 2020, reference 30 in the previous version of the manuscript)?

REVIEWER COMMENTS

Reviewer #1 (Remarks to the Author):

I have concerns about the statistical analyses.

Why was a Mann Whitney U test used for Fig. 2k? Students' T tests were used to analyze the same types of data in Fig. 2f, g, h, i, j and in Fig. 1g, h, i.

- We thank the reviewer for pointing out this issue. Indeed, we had made a mistake when analysing normality of our dataset; we have now redone the analyses and, since our data follow a normal distribution, all of them have been analysed by t-test, and Welch's correction has been applied for those datasets that were not homoscedastic (new Figure 2k). Details of the significance analysis for all datasets are shown below.

Figure 1.

	Fig. 1g	Fig. 1h	Fig. 1i
Table Analyzed	TRV-POLA2 Local	TRV-POLA2 Systemic	TRV-POLA1 Local
Test	Unpaired t test with Welch's correction	Unpaired t test with Welch's correction	Unpaired t test with Welch's correction
P value	0.0007	0.0004	0.0025
P value summary	***	***	**
Significantly different (P < 0.05)?	Yes	Yes	Yes
F test to compare variances			
P value	0.0280	<0.0001	<0.0001
P value summary	*	****	****
Significantly different (P < 0.05)?	Yes	Yes	Yes

Figure 2.

	Fig. 2f	Fig. 2g	Fig. 2h	Fig. 2i	Fig. 2j	Fig. 2k
Table Analyzed	TRV-POLD2 Local	TRV-POLE2 Local	TRV-POLE2 Systemic	TRV-POLD1 Local	TRV-POLE1 Local	TRV-POLE1 Systemic
Test	Unpaired t test with Welch's correction	Unpaired t test	Unpaired t test	Unpaired t test with Welch's correction	Unpaired t test	Unpaired t test with Welch's correction
P value	0.0001	0.0024	0.0001	0.0008	<0.0001	0.0964
P value summary	***	**	***	***	****	ns
Significantly different (P < 0.05)?	Yes	Yes	Yes	Yes	Yes	No
F test to compare variances						
P value	<0.0001	0.0977	0.3357	<0.0001	0.0525	0.0002

P value summary	****	ns	ns	****	ns	***
Significantly different (P < 0.05)?	Yes	No	No	Yes	No	Yes

Figure 3b (please see response below).

CS accumulation of TYLCV	
Welch's ANOVA test	
W (DFn, DFd)	51.29 (3.000, 9.578)
P value	<0.0001
P value summary	****
Significant diff. among means (P < 0.05)?	Yes

Games-Howell's multiple comparisons test	Significant?	Summary	Adjusted P Value
TRV-EV vs. TRV-POLA2	Yes	***	0.0008
TRV-EV vs. TRV-POLD2	No	ns	0.2268
TRV-EV vs. TRV-POLE2	Yes	**	0.0042
TRV-POLA2 vs. TRV-POLD2	Yes	***	0.0007
TRV-POLA2 vs. TRV-POLE2	Yes	**	0.0011
TRV-POLD2 vs. TRV-POLE2	Yes	**	0.0027

Figure 3c (please see response below).

CS accumulation of C3mut			
ANOVA summary			
F	32.27		
P value	<0.0001		
P value summary	****		
Significant diff. among means (P < 0.05)?	Yes		
R square	0.8288		
Bartlett's test			
Bartlett's statistic (corrected)	4.008		
P value	0.2606		
P value summary	ns		
Are SDs significantly different (P < 0.05)?	No		
ANOVA table	SS	DF	
Treatment (between columns)	16.76	3	MS
Residual (within columns)	3.462	20	5.586 F (DFn, DFd)
Total	20.22	23	0.1731 F (3, 20) = 32.27

Tukey's multiple comparisons test	Significant?	Summary	Adjusted P Value
TRV-EV vs. TRV-POLA2	Yes	****	<0.0001
TRV-EV vs. TRV-POLD2	Yes	*	0.0178
TRV-EV vs. TRV-POLE2	Yes	*	0.0101
TRV-POLA2 vs. TRV-POLD2	No	ns	0.1079
TRV-POLA2 vs. TRV-POLE2	Yes	****	<0.0001
TRV-POLD2 vs. TRV-POLE2	Yes	****	<0.0001

Figure 3d (please see response below).

VS accumulation of TYLCV	
Welch's ANOVA test	
W (DFn, DFd)	24.59 (3.000, 8.510)
P value	0.0002
P value summary	***
Significant diff. among means (P < 0.05)?	Yes

Games-Howell's multiple comparisons test	Significant?	Summary	Adjusted P Value
TRV-EV vs. TRV-POLA2	Yes	***	0.0009
TRV-EV vs. TRV-POLD2	Yes	**	0.0018
TRV-EV vs. TRV-POLE2	No	ns	0.0670
TRV-POLA2 vs. TRV-POLD2	No	ns	0.7996
TRV-POLA2 vs. TRV-POLE2	Yes	*	0.0215
TRV-POLD2 vs. TRV-POLE2	Yes	*	0.0203

Figure 3e (please see response below).

VS accumulation of C3mut	
Welch's ANOVA test	
W (DFn, DFd)	12.43 (3.000, 9.126)
P value	0.0014
P value summary	**
Significant diff. among means (P < 0.05)?	Yes

Games-Howell's multiple comparisons test	Significant?	Summary	Adjusted P Value
TRV-EV vs. TRV-POLA2	Yes	*	0.0290
TRV-EV vs. TRV-POLD2	Yes	*	0.0249
TRV-EV vs. TRV-POLE2	No	ns	0.1308
TRV-POLA2 vs. TRV-POLD2	No	ns	0.9564
TRV-POLA2 vs. TRV-POLE2	Yes	*	0.0158
TRV-POLD2 vs. TRV-POLE2	Yes	*	0.0148

Figure 4.

	Fig. 4c	Fig. 4d
Table Analyzed	TRV-POLE2 Local	TRV-POLE1 Local
Test	Unpaired t test with Welch's correction	Unpaired t test with Welch's correction
P value	0.0009	0.0127
P value summary	***	*
Significantly different (P < 0.05)?	Yes	Yes
F test to compare variances		
P value	<0.0001	0.0010
P value summary	****	**
Significantly different (P < 0.05)?	Yes	Yes

Supplementary Figure 6

	Supplementary Figure 6a	Supplementary Figure 6b	Supplementary Figure 6c
Table Analyzed	TRV-POLA2 Local	TRV-POLD2 Local	TRV-POLE2 Systemic
Test	Unpaired t test with Welch's correction	Unpaired t test with Welch's correction	Unpaired t test with Welch's correction
P value	0.0005	<0.0004	0.0363
P value summary	***	***	*
Significantly different (P < 0.05)?	Yes	Yes	Yes
F test to compare variances			
P value	0.0025	<0.0001	0.0112
P value summary	**	****	*
Significantly different (P < 0.05)?	Yes	No	Yes

As performed, the ANOVA analysis in Fig. 3 does not support the proposed differential roles of POLA and POLD in complementary and virion sense replication. In panel B, There are no statistical differences between complementary sense DNA levels for TYLCV-wt in the EV, POLA2, and POLD2 treatments. The same is true for TYLCV-C3. The only statistical difference is between EV (TYLCV-wt) and POLA2 (TYLCV-C3). In panel C, there were no statistical differences between between the virion sense DNA levels for the EV, POLA2, and POLD treatments for the wild-type and mutant viruses. The ANOVA analysis supports the conclusion that silencing POLE2 increase TYLCV-wt accumulation.

The authors did not provide information about their ANOVA model but it seems that they did a single analysis for each panel. If that is the case, a better approach might be to analyze the TYLCV-wt and TYLCV-C3mut results separately or include an interaction term in their model for the two viruses.

- **To compare groups following normality and homoscedasticity, we performed one-way ANOVA followed by Tukey's multiple comparisons test; when variances were not equal, we performed one-way ANOVA-Welch followed by Games-Howell's multiple comparisons test.**
Again, we have to apologize, since we had mistakenly uploaded the wrong version of this figure panel; the correctly analysed version is shown below.

Correct version:

We nevertheless agree with the reviewer in that separately analysing TYLCV WT and the C3 mutant is a better option, so we have now represented the corresponding results in two different panels (new Figure 3b-e; see below) and analysed the data in each of them by one-way ANOVA, as indicated in the new figure legend. We thank the reviewer for this suggestion.

Reviewer #2 (Remarks to the Author):

This is a revised manuscript that has taken into account previous reviewer comments. My initial comments have been addressed satisfactorily and the revisions based on other reviewer comments has made the manuscript much stronger.

- **We thank the reviewer for his/her positive assessment of our manuscript.**

Reviewer #3 (Remarks to the Author):

The authors have improved the manuscript.

The authors show now that silencing of the catalytic subunits of DNA POLA, POLD and POLE, namely POLA1, POLD1 and POLE1, results in the same effect as silencing of the regulatory subunits. In both cases silencing of the respective subunits leads to a decrease of viral DNA accumulation for POLA and POLD whereas silencing of POLE subunits results in an increase of viral DNA.

The quality of the blots has been improved. Nevertheless, some aspects are still not resolved. In the co-IP experiment, the size of NbPOLE2-RFP in the input fraction doesn't match the expected size of 63 kDa, but rather seems to be in the range of 55 kDa (or even below if taking the source data file with the molecular weight marker into account). The observed lower molecular weight might be the result of a truncation of the fusion protein which might in turn influence the interaction potential of POLE2 with C3.

Expression of the proteins used in BiFCs assays is now included in the Supplementary Information. Here again, it seems there are some inconsistencies. NbPOLD2-YC has an expected size of 28 kDa which is probably not the strong band on the blot around 40 kDa. For the same reason, the band for NbPOLE2-YC (expected size of 44 kDa) is not convincing. Together with the result of the co-IP it is still unclear if POLE2 and C3 are not interacting.

- **We thank the reviewer for raising this issue. Indeed, the size of the detected bands does not correspond to the theoretically predicted size; this phenomenon is frequently observed, and might be due to post-translational modifications or to inaccuracies in the annotation. Given that we are using cDNA cloned from *N. benthamiana*, we would expect these differences to similarly occur with the endogenous transcript. Nevertheless, we completely agree with the reviewer in that a negative result in these experiments cannot be taken for lack of real interaction, and so we have modified Line 119 accordingly: it now reads "...no interaction could be detected between C3 and the DNA polymerase ϵ regulatory subunit POLE2/DPB2".**

We would like to note that this undetected interaction is only mentioned at this point in the text, and it does not affect later results, interpretations, or the final model.

The discussion has improved but still needs language polishing (e.g. lines 207-209, 226/227).

- The sentences indicated by the reviewer are the following:
 1. “...whether DNA polymerase α performs the synthesis of the viral CS alone, despite its basal low processivity, or it does so in conjunction with a yet-to-be-identified polymerase, remains to be determined.”
 2. “Also worth considering is the fact that viral replication-related proteins are not present during the first phase of infection, ...”

We apologize but we fail to detect any problem in these sentences, hence we would appreciate if the reviewer could point out the specific issue he/she is referring to so that we can make the appropriate modifications.

Specific comments:

- Line 94: What is meant with thicker leaves? Is the whole leave (leave blade) thicker or the veins? It is not clear from the images in Supplementary Figure 4

- In these plants, the entire leaf blade is thicker – this is now specified in the text (line 92).

- I still didn't find a description for TRV-mediated gene-silencing in reference 39 (Medina-Puche et al. 2020, reference 30 in the previous version of the manuscript)?

- In Medina-Puche *et al.*, 2020, Figure 2L includes TRV-mediated silencing experiments in *N. benthamiana* and in tomato; the experimental details can be found in STAR Methods, e8. The VIGS assays in this work have been done following the procedure described therein.

REVIEWERS' COMMENTS

Reviewer #1 (Remarks to the Author):

The authors have redone some of the statistical analysis. It now supports the conclusions of the manuscript.

Reviewer #3 (Remarks to the Author):

The authors have improved the manuscript and have taken into account most previous reviewer comment.

- We thank the reviewer for raising this issue. Indeed, the size of the detected bands does not correspond to the theoretically predicted size; this phenomenon is frequently observed, and might be due to post-translational modifications or to inaccuracies in the annotation. Given that we are using cDNA cloned from *N. benthamiana*, we would expect these differences to similarly occur with the endogenous transcript. Nevertheless, we completely agree with the reviewer in that a negative result in these experiments cannot be taken for lack of real interaction, and so we have modified Line 119 accordingly: it now reads "...no interaction could be detected between C3 and the DNA polymerase ϵ regulatory subunit POLE2/DPB2"

The explanation for the discrepancy between expected sizes and observed sizes of various fusion proteins is very general, especially since some of the differences are quite large. It might be that this aspect cannot be clarified satisfyingly at this stage of the analyses.

- The sentences indicated by the reviewer are the following:

1. "...whether DNA polymerase α performs the synthesis of the viral CS alone, despite its basal low processivity, or it does so in conjunction with a yet-to-be-identified polymerase, remains to be determined."
2. "Also worth considering is the fact that viral replication-related proteins are not present during the first phase of infection, ..."

We apologize but we fail to detect any problem in these sentences, hence we would appreciate if the reviewer could point out the specific issue he/she is referring to so that we can make the appropriate modifications.

For the first example one suggestion for an easier reading would be:

DNA polymerase α , but not DNA polymerase δ , is essential for the accumulation of the dsDNA replicative intermediate. Despite its basal low processivity it remains to be determined whether DNA polymerase α performs synthesis of the viral CS alone or in conjunction with a yet-to-be-identified polymerase.

The second example is not well connected to the sentences before in my eyes and might need improvement.

Specific comments:

- In these plants, the entire leaf blade is thicker – this is now specified in the text (line 92).

Thank you for clarifying.

- In Medina-Puche et al., 2020, Figure 2L includes TRV-mediated silencing experiments in *N. benthamiana* and in tomato; the experimental details can be found in STAR Methods, e8. The VIGS assays in this work have been done following the procedure described therein.

Thank you. This might have been a problem with the versions: in the original version of the manuscript this reference was pointing to a bioRxiv article, which is now published in Cell.

Response to reviewers

We would like to thank all three reviewers for their careful assessment of our manuscript and their suggestions, which have undoubtedly contributed to improving this work.

Reviewer #3 (Remarks to the Author):

For the first example one suggestion for an easier reading would be:

DNA polymerase α , but not DNA polymerase δ , is essential for the accumulation of the dsDNA replicative intermediate. Despite its basal low processivity it remains to be determined whether DNA polymerase α performs synthesis of the viral CS alone or in conjunction with a yet-to-be-identified polymerase.

The second example is not well connected to the sentences before in my eyes and might need improvement.

> We have now rephrased the first sentence as indicated by the reviewer (Lines 202-205).

The other example indicated by the reviewer corresponds to the second of two sentences at the end of a paragraph in the Discussion section where we raise two aspects that need to be taken into account when considering the model proposed. We have now modified this sentence to read "It also needs to be considered that viral replication-related proteins are not present during the first phase of infection, hence the initial recruitment of DNA polymerase α and any other factors required for the synthesis of the dsDNA must happen without the contribution of these viral effectors."